# Fabrication of a Fish-Bone-Inspired Inorganic–Organic Composite Membrane

**DOI:** 10.3390/polym15204190

**Published:** 2023-10-23

**Authors:** YuYang Jiao, Masahiro Okada, Bhingaradiya Nutan, Noriyuki Nagaoka, Ahmad Bikharudin, Randa Musa, Takuya Matsumoto

**Affiliations:** 1Department of Biomaterials, Graduate School of Medicine, Dentistry and Pharmaceutical Science, Okayama University, 2-5-1 Shikata-cho, Kita-ku, Okayama 700-8558, Japan; phue18vu@s.okayama-u.ac.jp (Y.J.); m_okada@cc.okayama-u.ac.jp (M.O.); pcdu2sgb@s.okayama-u.ac.jp (B.N.); pkcf3z4e@s.okayama-u.ac.jp (A.B.); randaadilmoh@gmail.com (R.M.); 2Advanced Research Center for Oral and Craniofacial Sciences, Graduate School of Medicine, Dentistry and Pharmaceutical Sciences, Okayama University, 2-5-1 Shikata-cho, Kita-ku, Okayama 700-8558, Japan; nagaoka@okayama-u.ac.jp

**Keywords:** fish bone, lamellar structure, self-assembly, guided bone regeneration

## Abstract

Biological materials have properties like great strength and flexibility that are not present in synthetic materials. Using the ribs of crucian carp as a reference, we investigated the mechanisms behind the high mechanical properties of this rib bone, and found highly oriented layers of calcium phosphate (CaP) and collagen fibers. To fabricate a fish-rib-bone-mimicking membrane with similar structure and mechanical properties, this study involves (1) the rapid synthesis of plate-like CaP crystals, (2) the layering of CaP–gelatin hydrogels by gradual drying, and (3) controlling the shape of composite membranes using porous gypsum molds. Finally, as a result of optimizing the compositional ratio of CaP filler and gelatin hydrogel, a CaP filler content of 40% provided the optimal mechanical properties of toughness and stiffness similar to fish bone. Due to the rigidity, flexibility, and ease of shape control of the composite membrane materials, this membrane could be applied as a guided bone regeneration (GBR) membrane.

## 1. Introduction

Clinical researchers have long been interested in the best approaches to treat bone deformities caused by edentulism, trauma, and malignant tumors. Guided bone regeneration (GBR) membranes have been used in a variety of surgical procedures, including bone augmentation, periodontal flap surgery, and peri-implant bone-loss treatment [1]. GBR membrane not only promotes the proliferation of osteoprogenitor cells to generate new bone tissue, but also plays an important role in preventing the invasion of surrounding epithelial and connective tissue cells [2]. In addition to biocompatibility, an ideal GBR membrane must also satisfy other properties, such as mechanical properties. During tissue repair in vivo, membrane materials must be strong enough to withstand cellular traction and wound contraction forces. It is necessary to withstand external forces caused by the deformation of the body. A balance must be struck between the rate of membrane degradation and regeneration. Furthermore, the material should not be too difficult to handle during surgery [3]. Researchers have attempted to create a GBR membrane for therapeutic application using synthetic materials (such as PTFE (poly(1,1,2,2-tetrafluoroethylene)) [4]. Currently, membranes are mainly made of naturally derived polymers (collagen) or degradable synthetic materials such as PLGA (Poly(lactic-co-glycolic acid), which offer advantages compared to non-degradable membranes, such as avoiding secondary surgery [5]. Improvements are needed, since the membranes that are currently in use are difficult to handle during surgery and have mechanical properties that are very different from those of natural bone.

The development of materials with novel functions inspired by plants and other living things has gained interests in recent years for biomimetic materials [6,7,8]. Typical examples include adhesives that mimic gecko foot and superhydrophobic materials that mimic lotus leaves [9,10]. Natural biological materials that exhibit special properties are not found in artificial materials. When considering applications to bone-tissue-related materials, it may be possible to develop materials that combine strength, toughness, and flexibility, which are seemingly contradictory properties, by utilizing this biomimetic concept [11,12]. Materials that actually imitate natural bone tissues have been reported. For example, an inorganic–polymer scaffold was synthesized to mimic the nano-structural properties of native bone tissue [13]. In addition, a hybrid fibrous scaffold that closely mimics the bone extracellular matrix (ECM) was introduced to provide a host-friendly interface [14]. In these reports, knowledge about the structure and components or ideas about bone-mimetic materials are derived from mammalian bones [15,16]. On the other hand, the possibilities for novel bone-related materials will be substantially expanded by the addition of emerging knowledge about the bones of other organisms in addition to mammals. From this point of view, various bones such as intermuscular bones (IBs), vertebrae, and ribs have been investigated in the study of bones of teleost fish, which account for more than half of all extant vertebrate species [16,17,18,19]. In particular, fish ribs exhibit excellent mechanical properties and elastic bending forces, making them effective targets for bone-mimetic materials [20,21]. These higher mechanical properties are due to an inorganic–organic composite consisting of calcium phosphates (CaP) and collagen fibers. However, there are still unknowns regarding the mechanisms by which this property is expressed [22,23].

Herein, based on this background, we report a cost-effective fabrication technique for a fish-bone-like composite by fabricating an inorganic–organic composite membrane using a water-evaporation-induced assembly approach. First, we examined the microstructure and orientation of crucian carp fish ribs from a material science perspective. Furthermore, based on the results obtained here, we devised a new technique for the fabrication of layered inorganic–organic composite membranes and evaluated them. The proposed fabrication procedure takes advantage of the rapid formation of plate-like crystal brushite (dicalcium phosphate dihydrate, CaHPO_4_·2H_2_O) through a water/ethanol system. In the bottom-up method, the formation of the lamellar structure was facilitated by gentle drying caused by the gradual evaporation of water. By further optimizing the compositional ratio of the CaP filler and gelatin hydrogel, it was discovered that a 40% CaP filler content yields higher toughness and rigidity than fish bone. In addition, the membrane shape was easily controlled via the casting of a CaP–gelatin hydrogel sheet on a porous gypsum mold, and it would be useful as a cell barrier membrane. For the guided bone (or tissue) regeneration, the sheet should have patient-specific shape tunability for better application.

## 2. Materials and Methods

### 2.1. Characterization of Fish Bone

Twenty adult Carassius langsdorfii fish (Kawamura tansuigyo, Tokyo, Japan) with an average length of 72 ± 8 mm were utilized for this study. The rib bones on both sides were meticulously detached from the fish column, and any surplus connective tissue surrounding the isolated bones was eliminated using a dissection microscope. Samples were treated with a 4% paraformaldehyde (PFA) solution at a temperature of 4 °C for 48 h. Subsequently, the fixatives underwent washing 3 times in phosphate-buffered saline (PBS, pH 7.4) and were subjected to a 5-day decalcification process using a 0.5 M ethylenediaminetetraacetic acid (EDTA) solution. The samples were subjected to fixation using a 2% solution of OsO_4_ (osmium) for 24 h, followed by a gradient dehydration process using ethanol ranging from 70% to 100%. The samples were subsequently immersed in EPON 812 resin (TAAB Laboratories Equipment) for a duration of 3 days. Afterwards, they were sectioned into slices with a thickness of 80 nm for inspection under a Scanning Electron Microscope (SEM) (JSM-6701F, JEOL, Tokyo, Japan).

### 2.2. Fabrication of the CaP–Gelatin Lamellar Membrane

The solutions of 0.1M Ca(NO_3_)_2_·4H_2_O and 0.1M (NH_4_)_2_HPO_4_ were prepared by separately dissolving in ethanol/water mixture. Calcium phosphate (CaP) was synthesized by a rapid mixing of 0.1M Ca(NO_3_)_2_·4H_2_O and 0.1M (NH_4_)_2_HPO_4_ solutions (FUJIFILM Wako Pure Chemical, Tokyo, Japan) and stirred for 3 h. After filtration, the white precipitates were collected and dried for 12 h at 40 °C. Appropriate amounts of CaP powders were mixed with gelatin solution (Nitta Gelatin, Osaka, Japan) at 40 °C and stirred for several minutes. The initial quantities of the reagents were changed in accordance with the specified CaP–gelatin weight ratios, denoted as 0/100, 20/80, 40/60, 60/40, and 80/20. The mixed dispersion was subsequently poured into a stainless steel mold (SUS430, Trusco Nakayama, Tokyo, Japan) which was cooled at 0 °C to facilitate hydrogel formation and then gentle drying was conducted at 4 °C for 48 h. Meanwhile, an alternative sample was produced through the process of cooling the dispersion to 0 °C, followed by the application of rapid drying under vacuum conditions at room temperature (27 °C) for 2 h (Figure 1).

### 2.3. Physico-Chemical and Mechanical Characteristics of the CaP–Gelatin Composite

An X-ray diffraction (XRD) analysis (CuKα, 40 kV, and 200 mA, RINT2500HF, Rigaku, Tokyo, Japan) was carried out to identify the obtained crystals of the CaP–gelatin composite membrane at a scanning speed of 1° per minute from 15° to 60°. The relative intensity of the diffraction peaks (020), (121), and (112) in the XRD profile was used to evaluate the degree of preferential orientation of CaP crystals. The attenuated total reflectance Fourier transform infrared spectroscopy (ATR-FTIR) spectra (IR Affinity-1S, Shimadzu, Kyoto, Japan) were recorded after pressing the samples directly onto a ZnSe Prism at room temperature. The result was analyzed using spectrum analysis software (LabSolutions IR v2.13; Shimadzu, Kyoto, Japan). The microstructures of all samples were observed with SEM (S-4800, Hitachi, Ibaraki, Japan) at an acceleration of 5 kV. To obtain high-resolution images of the elements present in the samples, backscattered electrons were used. The mechanical properties of the CaP–gelatin composite membrane were tested with a three-point bending test (EZ-SX500N, Shimadzu). The investigation of the membrane’s swelling capability involved immersing the sample in distilled water at a temperature of 37 °C for 3 h (Appendix A). The magnitude of swelling was assessed by employing the subsequent mathematical expression.
(1)$$Swelling\;\left(\%,\frac{w}{v}\right)=\frac{\left(W_{t}-W_{0}\right)}{W_{0}}\times 100$$
where *W*_0_ is weight of the CaP–gelatin composite membrane before swelling and *W_t_* is the weight of the CaP–gelatin composite membrane after swelling.

### 2.4. Shape Control of Fish Bone Biomimetic Membrane by Slip Cast on Gypsum

For the clinical implementation of this membrane material, it is necessary to adapt the membrane’s shape to the bone structure of each patient’s surgical region. Therefore, the formability of the composite membrane was investigated using a model of an edentulous human mandible. The CaP–gelatin composite membrane was dipped in water until it became flexible, and then it was set on the gypsum surface with variable sizes. The gypsum mold designs were fabricated based on 3D design with different sizes, which were regarded as a control group. The shape control of the CaP–gelatin membrane was carried out on a porous gypsum mold synthesized by combing water with gypsum powders (premium dental gypsum, Premium Plus, Tokyo, Japan) at a water-to-gypsum ratio of 0.4. The curvature (K) of a composite material is used to describe how precisely the shape can be controlled. The curvature of both the CaP–gelatin membrane and the 3D design model were calculated by using following formula. The calculation results were marked as theoretical and actual calculation, respectively.
(2)$$K=\frac{1}{R}\;,\;\;\;\;R=\frac{\left(a^{2}+h^{2}\right)}{2h}\;$$
where *K* is the curvature of the composite, *R* is the radius of the composite, *a* is the half chord length of the curved composite, and *h* is the distance of the arc top to chord length; based on Pythagoras’ theorem, *R* can be calculated.

## 3. Results

### 3.1. Fish Bone Microstructure Observation

The SEM observation of the microstructure of ribs embedded in resin revealed that there was tissue occupied by cellular substances in the center of the ribs, and the surrounding area was covered with calcified tissue. Osteocytes were observed in this calcified area, and these osteocytes were oriented in the same direction as the long axis of the ribs. A more enlarged photograph of the undecalcified section revealed a bone matrix with a special orientation consistent with the long axis of the rib. This bone matrix was shown to exhibit a layered structure (Figure 2A–D). As a result of observation using tissue decalcified with EDTA, the presence of fibers oriented in the same direction as the long axis direction was observed. Further magnification showed that the fibers were collagen, due to the bundle structure observed (Figure 2E,F).

### 3.2. Morphology Control of Calcium Phosphate Particles

CaP particles were synthesized in the mixed solution of water (W) and ethanol (E). In brief, when the ratio between W and E was 100:0 (W100E0), small cubic-like nanoparticles were formed (Figure 3A). With increase in ethanol fractions from 0 to 70 wt% (W70E30, W50E50, and W30E70), a transition from a three-dimensional irregular shape to a one-dimensional growth of crystals with regular shape was corroborated, and consequently plate-like crystals were obtained (Figure 3B–D). Due to the insolubility of (NH_4_)_2_HPO_4_ in ethanol containing more than 70 wt%, no further CaP particles were synthesized in this study. The crystals’ thickness decreased from 0.48 ± 0.1 to 0.41 ± 0.2 μm and the length increased from 26.3 ± 0.3 to 34.24 ± 0.6 μm, according to the Image J measurement (Figure 3E). Consequently, by increasing the proportion of ethanol, longer and thinner platelet crystals were obtained by calculating the aspect ratio, which is calculated by dividing the length by the thickness (Figure 3F). Meanwhile, the identification was further confirmed by XRD analysis, with major peaks identified at 2θ° = 11.8°, 21.1°, and 29.5°, corresponding to the formation of brushite (CaHPO_4_·2H_2_O), (020), (121), and (112) crystal planes, respectively (Figure 3G). To fabricate a membrane with a layered structure, plate-like CaP was used as a raw material in the further study.

### 3.3. Fabrication of a CaP–Gelatin Membrane with Lamellar Structure

An inorganic–organic composite membrane was obtained through the gradual drying of CaP–gelatin hydrogel for extended hours. Under this fabrication condition, no phase change in the inorganic crystal was detected, as observed in the XRD pattern (Figure 4A). Following the coating of CaP platelets with gelatin, the emerging FTIR peak of the C-N-H and COO^−^ bands of the CaP–gelatin composite showed that gelatin molecules were absorbed on the surface of the CaP platelets as a result of the Ca^2+^-coordinated ionic interactions between the gelatin molecules and the inorganic platelets (Figure 4B). Meanwhile, the influence of the fabrication condition on the microstructure of composite membrane was examined using SEM. The layered structure changed depending on the drying conditions of the membrane. When rapid drying was performed, the inorganic crystallites of the membrane exhibited random orientation (Figure 4C). On the other hand, the inorganic crystals exhibited an ordered layered structure via gentle drying (Figure 4D,E). Furthermore, even in the case of mild synthesis, when the amount of CaP added was increased (more than 40%), it was observed that the layered structure was disturbed and voids were formed between CaP particles (Figure 4F).

### 3.4. Investigation of Mechanical Properties of the Composite Membrane

The mechanical properties of the fish bone biomimetic composite membrane were systematically studied and compared to those of the natural fish bone to prove the validity of our fish bone biomimetic design. Despite a small decrease in stiffness, the ultimate flexural strength of the composite membrane reached 94.0 MPa, which is higher than that of native fish rib bone (80.1 MPa) and pure gelatin bulk (47.0 MPa) (Figure 5A). On the other hand, simply incorporating CaP powders into the gelatin matrix without structural control can only improve the stiffness of the composite (Figure 5B). The mechanical properties of the composite membrane were further optimized by adjusting the mass ratio between CaP powders and gelatin constituents. The yield strength enhanced notably with the CaP content. The optimal mechanical strength was determined to contain 40 wt% of CaP platelets, corresponding to the maximum Young’s modulus and toughness of 1.31 GPa and 6.66 J/m^3^. The composites failed at obviously lower strains and the strength abruptly decreased as CaP content surpassed 40 wt% (Figure 5C). For a pure CaP, no data could be obtained because it was too brittle to be tensile tested.

### 3.5. Shape Control of CaP–Gelatin Membrane

The CaP–gelatin composite membrane reswollen with water was soft and could be reformed on the jawbone porous gypsum cast (Figure 6A). The membrane was dried on gypsum at 80 °C for 6 h, resulting in a three-dimensional membrane that reflected the jawbone model shape (Figure 6B). The resulting shape-controlled composite membrane maintained its pearlescent color (Figure 6C). Various sizes of gypsum molds and CaP−gelatin membrane shapes could be precisely controlled. The curvature of the fabricated composite membrane with different sizes was calculated. Here, we confirmed a good match between theoretical and actual calculation when the radius was in large range. In contrast, when the radius of the membrane gradually became smaller, less than 2.5 mm, the error bar size increased. Since the size requirement of regular GBR membrane is more than 3.0 mm, this indicated that we could successfully reproduce the membrane using gypsum in detail (Figure 6D–F).

## 4. Discussion

When designing biomimetic materials, tissues from various organisms can be used as references in order to obtain the desired properties. Recent studies have demonstrated that fish bones, including both osteocytic and anosteocytic types, have distinct properties with respect to their mechanical properties and composition. These attributes place fish bones in a distinct category within the bone material and distinguish them from bone materials found in other vertebrates [24]. In this research, we proceeded with the development of a new inorganic–organic composite membrane material with a layered structure based on fish ribs. 

Fish ribs are considered to be an excellent material with high toughness and rigidity [17,21,25]. To understand the mechanisms behind this characteristic statement, we focused on the microstructure of fish ribs. This microstructure can be broadly classified into two different parts: a central part and peripheral part. Unlike mammalian bone tissue, where the central region is mainly composed of bone marrow, fish bones have a significant population of chondrocyte-like cells, mainly in their central part [26]. In longitudinal sections, SEM observation reveals highly aligned minerals and collagen fibers in the same direction as the longitudinal axis [27]. These collagen bundles, with a small diameter and compact arrangement, are complexed with minerals to form a layered structure. This layered structure, composed of organic and inorganic materials, is thought to play an important role in increasing strength [28].

First, we investigated the shape control of the inorganic materials used. In order to produce an inorganic–organic composite material with a layered structure, it is desirable that the inorganic crystal has a plate-like structure [29]. In general, the growth of calcium phosphate-based inorganic crystals is greatly influenced by the environment during crystal growth, especially the ion concentration, pH, and hydration state [30]. The water/ethanol mixture plays an important role in controlling the morphological structure of CaP particles. In detail, the addition of ethanol (0–70 wt%) gradually weakens the hydrogen bonds between water molecules, reducing the energy required for dehydration and, as a result, breaking the interaction between calcium ions and water molecules [31]. As a result, more free calcium ions are released and start interacting with phosphate ions, leading to CaP formation [32,33]. As the number of CaP seeds increases within a certain spatial region, the formation of CaP structures of significant size begins, which can also be determined by XRD analysis. The presence of structured water during the process reduces the interfacial free energy and thus promotes the precipitation of CaP. Varying the ethanol-to-water ratio (W70E30, W50E50, W30E70) may provide further control over morphology [34,35]. This will lead to a significant increase in the nucleation rate, exceeding that of other thermodynamically favorable phases such as OCP and HAp [30]. This may elucidate the fundamental factors contributing to the change in CaP morphology from one-dimensional needle-like nanowires to three-dimensional plate-like shapes [36,37].

Next, we combined CaP platelets and gelatin to fabricate an inorganic–organic layered composite membrane. Here, it was assumed that plate-like crystals of CaP will be used, and gelatin will act as a binder. On the other hand, it is also known that gelatin itself is a material that exhibits high strength when dried [38]. Previous reports have shown that clay–gelatin composites exhibit a self-orientation of anisotropic nanoplatelets by controlling water evaporation during fabrication [39]. In this study, we confirmed that gentle drying and a high platelet-to-polymer ratio are essential to fabricate an inorganic–organic composite membrane with a lamellar structure similar to fish bones. During the drying process, the gradual evaporation of water does not completely destroy the network structure of the hydrogel [40]. As more water is gradually removed, the gelatin molecules bound to CaP platelets start to concentrate towards the bottom and naturally form a layered structure (Figure 7). Therefore, the tight adhesion between the inorganic components and their continuous stacking in this lamellar structure formation is induced by sufficient gelatin solution diffusion and interaction between the plate-like CaP, with the gradual evaporation of water [11]. On the other hand, the use of rapid drying techniques results in non-uniform gelatin solution diffusion between CaP and the development of disordered structures inside the inorganic–organic composite membrane (Figure 7) [11,41]. Furthermore, by changing the layered structure and composition of the composite material, researchers have succeeded in imparting both high mechanical strength and a certain level of toughness to the composite material [42]. This phenomenon can be explained by the interaction between CaP and gelatin molecules. Examining the chemical interaction between Ca^2+^ and COO^−^ detected by FT-IR, we found that when the CaP content is low, CaP can mostly contact gelatin molecules. As more CaP is added, less gelatin is available to interact with CaP, creating gaps between the inorganic crystals. The gaps increase depending on the amount of CaP, leading to a decrease in mechanical properties of both strength and toughness [43,44]. Furthermore, the shape of the membrane can be tailored to the application site. This is made possible by the swelling and softening of gelatin as it hydrates. As a result of this study, it was confirmed that this material has good handling characteristics.

Many studies have been conducted to date on the production of GBR membranes centered on collagen. Additionally, attempts have been made to add CaP to control strength. Song et al. indicated the potential utility of the collagen–apatite nanocomposite membrane in the domain of GBR by regulating the apatite particles size [45]. Another study also examined the potential application of collagen membranes with variable degrees of mineralized CaP [46]. However, compared to other studies, the major feature of this study is that the material was designed to imitate fish bones to increase strength. Therefore, we investigated the microstructure of fish rib bones from a material science perspective. Another important point of this study is that the layered structure was fabricated from an inorganic–organic composite. As a result, we succeeded in creating a membrane with both high mechanical properties and flexibility. In this way, understanding the fine structure of biological tissue, which has not been studied so far, from a material science perspective is effective in designing new materials. Moreover, this composite membrane, with excellent mechanical properties and morphology manipulation, would be a promising material for GBR applications. 

## 5. Conclusions

The fabrication of a lamellar membrane, which emulates the structure of fish rib bones, was undertaken by integrating plate-like calcium phosphate (CaP) crystals with gelatin as the organic component. This approach was motivated by the structural characteristics observed in fish rib bones. By manipulating the morphological shape of CaP in a water–ethanol system and adjusting the drying speed of the CaP–gelatin hydrogel system, a lamellar structure resembling fish bones was successfully obtained. Furthermore, the mechanical property was modified and reinforced by manipulating the platelet-to-polymer ratios. Moreover, the shape of this membrane composite could also be precisely regulated. Due to its superior mechanical properties and capacity for precise morphological control, this membrane is regarded as a promising material for guided bone regeneration (GBR) membranes.

## Figures and Tables

**Figure 1 polymers-15-04190-f001:**
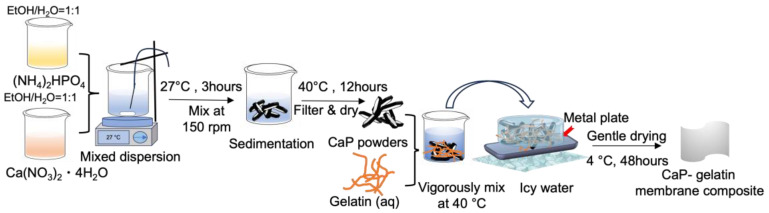
Schematic image of CaP synthesis and CaP–gelatin composite membrane preparation.

**Figure 2 polymers-15-04190-f002:**
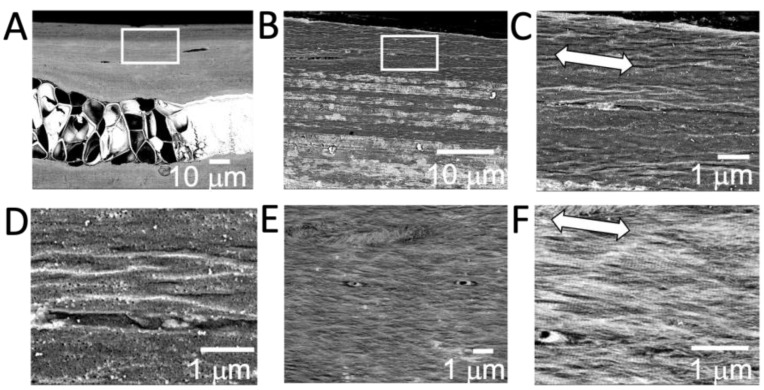
SEM observation of the cross section of fish rib bone. (**A**) Chondrocyte area can be found in the center of the fish rib bone. (**B**,**C**) Layered structures with highly oriented minerals are observed in the upper rectangular area. Arrow indicates the long axis direction of rib bone. (**D**) Images of decalcified fish rib bone from the upper side show the lamellar structure of the bone matrix. (**E**,**F**) Highly oriented collagen fibers can be observed. Arrow indicates the long axis direction of the rib bone.

**Figure 3 polymers-15-04190-f003:**
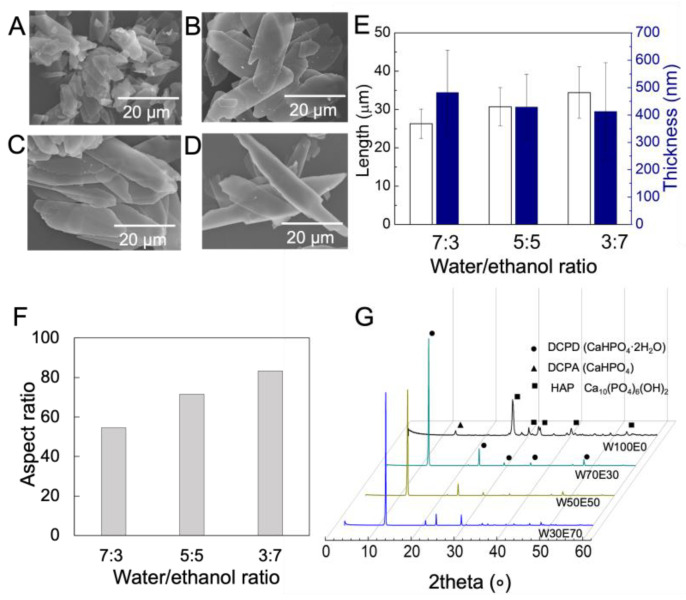
Characteristics of synthesized CaP particles. (**A**–**D**) SEM images show the synthetic CaP crystals with different sizes under various W/E volumetric ratios ((**A**) W100E0, (**B**) W70E30, (**C**) W50E50, (**D**) W30E70). (**E**,**F**) Plotted graphs show the average length, thickness, and aspect ratio of CaP crystals measured with image J. The white and blue colors represent the length and thickness of particles, respectively. (**G**) X-ray diffraction (XRD) pattern of CaP crystals fabricated under various W/E volumetric ratios.

**Figure 4 polymers-15-04190-f004:**
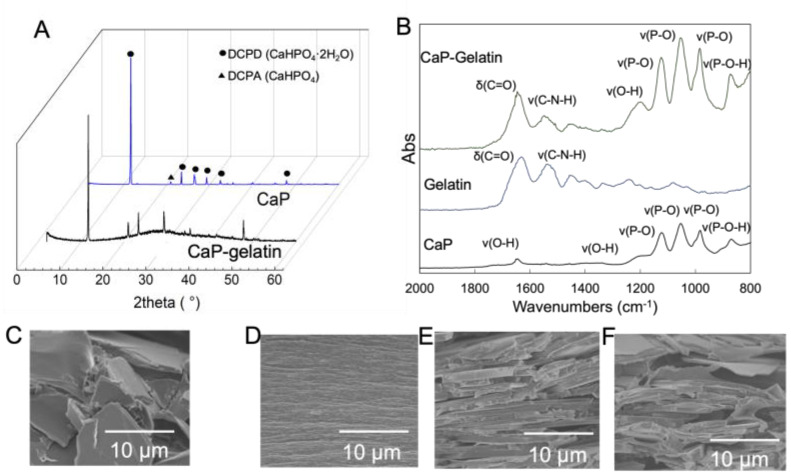
Characterization and microstructure of CaP–gelatin composite membranes. (**A**) X-ray diffraction (XRD) of CaP–gelatin composite and brushite. (**B**) ATR-FTIR of brushite, gelatin, and CaP–gelatin composite. (**C**) Cross-section of rapidly dried composite with lamellar structure containing different amounts of CaP ((**D**): 0 wt%; (**E**): 40 wt%; (**F**): 80 wt%).

**Figure 5 polymers-15-04190-f005:**
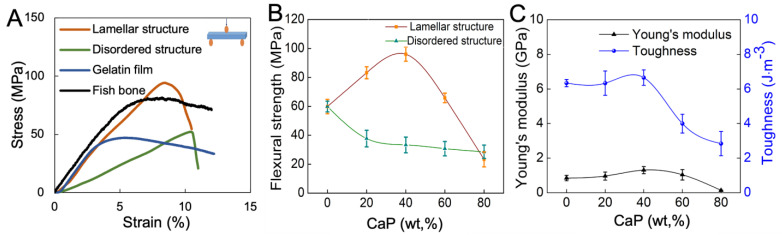
Mechanical properties of the fish bone biomimetic membranes containing different amounts of CaP platelets. (**A**) Comparison of stress–strain curves of the composite membranes between lamellar structure, disordered structure, gelatin membrane, and fish bone. (**B**) Comparison of mechanical strength between lamellar and disordered structures with different CaP content. (**C**) Mechanical properties of the Young’s modulus and toughness of composite membrane with different CaP content.

**Figure 6 polymers-15-04190-f006:**
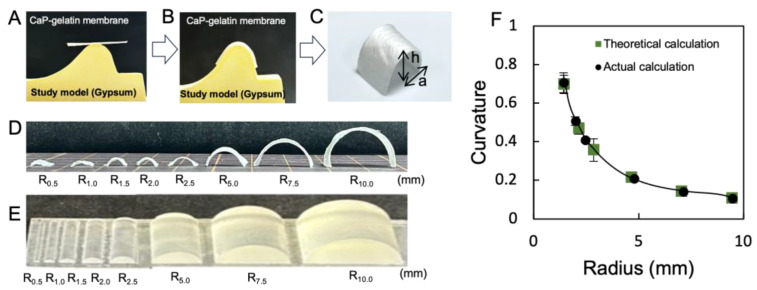
Shape control of CaP–gelatin membrane by gypsum. (**A**) Dried CaP–gelatin membrane attached onto the gypsum study model. (**B**) The swollen membrane was deformed according to the gypsum shape. (**C**) CaP–gelatin membrane can be obtained by reflecting the shape of the gypsum mold. (**D**) Different sizes of CaP–gelatin membrane can be made by placing on different-sized gypsum. (**E**) Three-dimensional design mold with different sizes (R = 0.5~10.0 mm). (**F**) Precise details can be replicated based on the curvature formula: a refers to the half-length of the chord and h is the distance from the arc top to chord length, and based on vertical theorem, R (radius) can be calculated.

**Figure 7 polymers-15-04190-f007:**
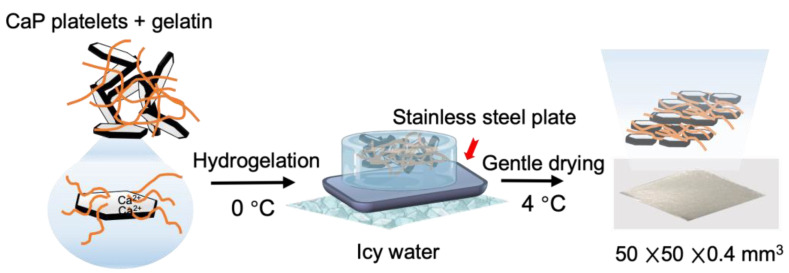
Lamellar structure was achieved through gentle drying of CaP–gelatin hydrogel at 4 °C for 48 h.

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
