# Peer review of "Fabrication of a Fish-Bone-Inspired Inorganic–Organic Composite Membrane"

_polymers, 2023, doi:10.3390/polym15204190_

Round 1
Reviewer 1 Report
Authors present a paper on the preparation and characterization of fish-bone-inspired organic-inorganic composite. The idea is very interesting and experiments seem to be correctly performed but the presentation requieres quite huge changes to be acceptable. Starting from the abstract, it should be rewritten (apart from improving english) in order to show only the most important results or findings, underlining them. It is not necessary to place conclusions, background, etc.
Moreover, figures do not present good quality. SEM images of figure 1 should be bigger and present better resolution for corraborating their findings. The same for that of figure 2. The errors in mechanical properties (ie deviations) are not clearly seen and should be improved as they become very important in order for result interpretation.
Conclusions should also be improved and completed with some strong sentences clearly showing the most important findings.
Finally, to employ the words "bone regeneration" in the tittle seems to be quite presumptuous as many tests and compatibilization issues should be done and probed before the use on this applications.
The quality is quite low, especially in the abstract, that should be rewritten. As basic grammar errors are detected, I strongly recommend an english native to correct the language.
Author Response
First of all, on behalf of my co-authors, I would like to thank you for the consideration given to re-submit our manuscript.
Secondly, I would like to express our sincere gratitude for the generous extension of the submission deadline.
Please find the detailed responses in attachment file.

Reviewer 2 Report
The authors have conducted research that could be of great interest to readership dealing with bone regeneration and composite materials. However, the article is not well structured and creates confusion. The following issues must be resolved before publishing:
1. Abstract must be rewritten, in order to provide concise and precise information about the research.
2. In the Introduction, include studies and results of collagen-HA used for the same purpose of bone regeneration; emphasize the advantages of your study
3. The sentence in Line 63 must be written in a different form.
4. In Subsection 2.2, include a Scheme representing the experimental preparation of membranes
5. Write XRD, FTIR, and SEM discussion separately, with a more thorough analysis and comparison with the literature.
6. The conclusion must be written with more details.
As mentioned to the authors, the sentence in Line 63 must be written in a different form.
Author Response
First of all, on behalf of my co-authors, I would like to thank you for the consideration given to re-submit our manuscript.
Secondly, I would like to express our sincere gratitude for the generous extension of the submission deadline.
Please find the detailed responses in the attachment.

Round 2
Reviewer 1 Report
In their new version, authors have addressed required questions and corrections, so the paper can be now published in its actual form.
Reviewer 2 Report
The authors have made corrections in accordance with the reviewer's suggestions.